# Constructing Co_3_O_4_ Nanowire@NiCo_2_O_4_ Nanosheet Hierarchical Array as Electrode Material for High-Performance Supercapacitor

**DOI:** 10.3390/nano14211703

**Published:** 2024-10-24

**Authors:** Bo Xu, Lu Pan, Yaqi Wang, Menglong Liu

**Affiliations:** School of Chemistry and Materials Engineering, Huainan Normal University, Huainan 232038, China; boxu6673561@sina.com (B.X.); 88wyq1219@sina.com (Y.W.);

**Keywords:** Co_3_O_4_ nanowire, NiCo_2_O_4_ nanosheets, composites, hierarchical array, supercapacitor

## Abstract

The Co_3_O_4_ nanowire@NiCo_2_O_4_ nanosheet hierarchical array was constructed on Ni foam using hydrothermal and annealing approaches in turn, from which a NiCo_2_O_4_ nanosheet could self-assemble on the Co_3_O_4_ nanowire. The structure and morphology of the Co_3_O_4_ nanowire@NiCo_2_O_4_ nanosheet hierarchical array were characterized via XRD, EDS, SEM, and FESEM, respectively. The electrochemical performance of the composite array was measured via a cyclic voltammetry curve, galvanostatic current charge–discharge, charge–discharge cycle, and electrochemical impedance and then compared with the Co_3_O_4_ nanowire. The results show that the Co_3_O_4_ nanowire@NiCo_2_O_4_ nanosheet hierarchical array could reach a high value of 2034 F g^−1^ at a current density of 2.5 A g^−1^. After 5000 galvanostatic charge–discharge cycles, the specific capacitance of the Co_3_O_4_ nanowire@NiCo_2_O_4_ nanosheet hierarchical array could still maintain 94.7% of the original value. Therefore, the Co_3_O_4_ nanowire@NiCo_2_O_4_ nanosheet hierarchical array would be a desirable electrode material for a high-performance supercapacitor.

## 1. Introduction

Co_3_O_4_ is widely regarded as a promising electrode material for pseudocapacitors due to its high theoretical specific capacitance, low consumption, and environmental friendliness [1,2,3,4,5,6]. Co_3_O_4_ nanomaterials with uniform size and special morphology, such as nanosheets, mesoporous nanotubes, and nanospheres, have been reported [7,8,9,10,11,12,13]. However, the specific capacitance, energy density, and cyclic performance of pure-phase Co_3_O_4_ nanoparticles are limited. The exploration of novel Co_3_O_4_-based nanocomposite electrode materials for supercapacitors have caused extensive research in recent years [14,15,16,17,18,19]. Co_3_O_4_-based nanocomposite electrode materials mainly include Co_3_O_4_@C, Co_3_O_4_/graphene, Co_3_O_4_@metal hydroxide, and Co_3_O_4_@metal oxide [20,21,22,23,24,25,26,27]. Although these Co_3_O_4_-based nanocomposite electrode materials showed improvement in the supercapacitor electrochemical performance compared with the Co_3_O_4_ nanostructures, the specific capacitance of these composite electrode materials is not very large, and the energy density is not high. It is still a great challenge to further design multi-component core–shell or heterogeneous electrode materials with higher electrochemical performance and study the corresponding relationship between structure and performance through simple synthesis. The s as-prepared NiCo_2_O_4_ with the similar spinel structure of Co_3_O_4_ has better electron conductivity, higher electrochemical activity, higher efficiency, lower consumption, and better environmental protection than the traditional transition metal group oxides [28,29,30,31,32,33]. Recently, an effective method of improving the electrochemical performance of the supercapacitor was to synthesize two metal oxides with similar structures on conductive templates, such as Co_3_O_4_@NiMoO_4_ [34,35], Co_3_O_4_/NiCo_2_O_4_ [36,37], and Co_3_O_4_@CoMn_2_O_4_ core-shell array structure [38].

In this work, the Co_3_O_4_ nanowire@NiCo_2_O_4_ nanosheets hierarchical array were constructed on a Ni foam substrate via the hydrothermal method, followed by the annealing process. The NiCo_2_O_4_ nanosheets are wrapped on the surface of the Co_3_O_4_ nanowire, which could provide faster ion and electron transfer and improve the structural integrity. The Co_3_O_4_ nanowire@NiCo_2_O_4_ nanosheets hierarchical array exhibited high specific capacitance and excellent cycling performance, which could be attributed to the synergic effect of the hierarchical nanostructure. The specific capacitance of the Co_3_O_4_ nanowire@NiCo_2_O_4_ nanosheet hierarchical array could reach 2034 F g^−1^ at a current density of 2.5 A g^−1^ and maintain 94.7% of the original value after 5000 charge–discharge cycles. The Co_3_O_4_ nanowire@NiCo_2_O_4_ nanosheet hierarchical array showed great potential application in supercapacitors.

## 2. Experimental Section

All chemical agents were of analytical grade and were used directly without further purification.

### 2.1. Synthesis of Co_3_O_4_ Nanowire

A total of 1.50 mmol of CoCl_2_·6H_2_O, 1.00 mmol of NH_4_F, and 2.00 mmol of urea were dissolved in 40 mL of distilled water under magnetic stirring. The obtained uniform solution was transferred into a 50 mL polytetrafluoroethylene reactor with 2 cm × 3 cm Ni foam and then hydrothermally treated at 120 °C for 6 h. The Ni foam was pre-treated in 6 M HCl. The obtained samples were washed with deionized water and anhydrous ethanol several times. Finally, the washed sample were dried at 60 °C in vacuum and then calcinated at 300 °C for 2 h with a heating rate of 5 °C min^−1^.

### 2.2. Synthesis of the Co_3_O_4_ Nanowire@NiCo_2_O_4_ Nanosheet Hierarchica Array

A total of 0.10 mmol NiCl_2_·6H_2_O, 0.20 mmol CoCl_2_·6H_2_O, 0.10 mmol NH_4_F, and 0.20 mmol urea were accurately weighed and resolved into the mixed solution of 30 mL methanol and 6 mL H_2_O; after stirring for 20 min, the uniform solution was transferred to a 50 mL polytetrafluoroethylene reactor with Ni foam-covered Co_3_O_4_ nanowire and hydrothermally treated at 120 °C for 6 h. The obtained samples were washed with deionized water and anhydrous ethanol several times. Finally, the precursors were dried in a vacuum at 60 °C for 10 h, and final samples were obtained by calcining the precursors at 300 °C for 2 h with a heating rate of 5 °C min^−1^.

### 2.3. Characterization of the Co_3_O_4_ Nanowire@NiCo_2_O_4_ Nanosheet Hierarchica Array

The phase purity of the samples were characterized using an XRD-6000 type X-ray powder diffraction instrument (Shimadzu Corporation, Kyoto, Japan) with a graphite monochromator and high-intensity Cu Ka radiation (λ = 0.154060 nm). The scan speed is 8° min^−1^. The scan range is 10° to 80°. The field emission scanning electron microscope (FESEM) images of the samples were observed using a Hitachi S-4800 (Tokyo, Japan) field emission scanning electron microscope (accelerated voltage: 5.0 kV). The transmission electron microscope (TEM) images of the samples were observed using a JEOL 2010 (Tokyo, Japan) transmission electron microscope (accelerated voltage: 200 kV).

### 2.4. Electrochemical Measurement of Samples

The supercapacitor electrochemical performance was measured using a CHI 660D electrochemical workstation (Shanghai Chenhua Instrument Co., Ltd., Shanghai, China). We cut the calcinated Ni foam covering the sample into small pieces of 1 cm × 1 cm. The 1 cm × 1 cm Ni foam covered the sample assembled with a platinum electrode and a standard saturated calomel electrode to form a three-electrode device with the platinum electrode as the counter electrode and the standard saturated calomel electrode as the reference electrode. The cyclic voltammetry curve, galvanostatic charge–discharge, cycling performance, and electrochemical impedance were measured with 3 M KOH solution as the electrolyte.

### 2.5. Electrochemical Calculation

The electrochemical performance of the specific capacitance Csp of the Co_3_O_4_ nanowire@NiCo_2_O_4_ nanosheet hierarchical array supercapacitor was calculated via the following formulae:$$C_{sp}=\frac{\int IdV}{S\times \Delta V\times m},$$
$$C_{sp}=\frac{I\times \Delta t}{m\times \Delta V},$$
where *I* is the current density of galvanostatic charge and discharge, Δt is the discharge time, m is the mass of the active materials, ΔV is the potential window, and *S* is the scan speed.

## 3. Results and Discussion

### Structure and Morphology of the Samples

The phase composition of the samples was tested on the X-ray powder diffraction instrument. The results are exhibited in Figure 1. The diffraction peaks of the Co_3_O_4_ nanowire include the (111), (220), (311), (222), (400), (422), (511), (400), (650), and (533) crystal phases, which can all be indexed to the characteristic peaks of Co_3_O_4_, indicating that the sample was cubic-phase Co_3_O_4_ (JCPDS:78-1969). Pure cubic-phase Co_3_O_4_ nanowire provided the theoretical basis for the preparation of the Co_3_O_4_ nanowire@NiCo_2_O_4_ nanosheet hierarchical array structure. Compared with the diffraction pattern of the Co_3_O_4_ nanowire, the diffraction peaks of the Co_3_O_4_ nanowire@NiCo_2_O_4_ nanosheet hierarchical array demonstrated basically no change except for a slightly enhanced diffraction peak intensity, which can be indexed to the characteristic peaks of spinel-phase NiCo_2_O_4_ (JCPDS:73-1702), indicating formation of the core–shell structure of the Co_3_O_4_ nanowire@NiCo_2_O_4_ nanosheet hierarchical array.

A field emission scanning electron microscope was used to observe the morphology and size of the samples. Figure 2a,b are the low-magnification and high-magnification SEM images of the Co_3_O_4_ nanowires. It can be seen that the as-prepared Co_3_O_4_ nanowire array is rather neat and uniform. Figure 2c,d are low-magnification and high-magnification SEM images of the Co_3_O_4_ nanowire@NiCo_2_O_4_ nanosheets hierarchical array. A flower-like nanosheet developed and was surrounded tightly by the Co_3_O_4_ nanowire, and the neat array was formed.

The morphology and structure of the samples were further analyzed via a transmission electron microscopy. As can be seen from Figure 3a–c, the constructed Co_3_O_4_ nanowire array and Co_3_O_4_ nanowire@NiCo_2_O_4_ nanosheet hierarchical array were all porous in structure, which would increase the number of electrochemical active sites [39,40]. The high-resolution lattice fringes of the samples are shown in Figure 3b–d. The crystal plane spacing of the Co_3_O_4_ nanowire array was 0.47 nm, corresponding to the (111) crystal plane. The crystal plane spacing of the Co_3_O_4_ nanowire@NiCo_2_O_4_ nanosheet hierarchical array was 0.245 nm, corresponding to the (111) and (311) crystal planes.

The EDS mapping images are shown in Figure 4. Figure 4a shows that there is 18.48 wt% of the Ni element. Figure 4b showed that the Ni element was uniformly distributed. It also confirmed that the NiCo_2_O_4_ nanosheet was successfully constructed on the Co_3_O_4_ nanowire.

The Co_3_O_4_ nanowire array and the Co_3_O_4_ nanowire@NiCo_2_O_4_ nanosheet hierarchical array were used as electrode materials to test the supercapacitor electrochemical performance. The results are shown in Figure 5. Cyclic voltammetry curves of the samples were tested under different scan speeds (from 5 to 50 mV s^−1^) in a voltage range from −0.1 to 0.5 V. The redox peaks in the cyclic voltammetry curves of the Co_3_O_4_ nanowire electrode material (Figure 5a) are due to the Co^2+^/Co^3+^ and Co^3+^/Co^4+^ redox reactions [41,42] (Equations (1) and (2)):
Co_3_O_4_ + OH^−^ + H_2_O = 3CoOOH + e^−^,(1)
CoOOH + OH^−^ = CoO_2_ + H_2_O + e^−^.(2)

For the Co_3_O_4_ nanowire@NiCo_2_O_4_ nanosheet hierarchical array, the expansion and migration of the redox peak is mainly due to the increase in the Ni^2+^/Ni^3+^ reaction (Figure 5c) [37] (Equation (3)):
NiCo_2_O_4_ + OH^−^ + H_2_O = NiOOH + 2CoOOH + e^−^.(3)

According to the above supercapacitor electrochemical calculation (Equation (1)), the specific capacitance values were calculated at different scan speeds. At the scan speeds of 5 mV s^−1^, 10 mV s^−1^, 25 mV s^−1^, and 50 mV s^−1^, the specific capacitance values of the Co_3_O_4_ nanowire array were 1267 F g^−1^, 1108 F g^−1^, 941 F g^−1^, and 715 F g^−1^; and the specific capacitance values of the Co_3_O_4_ nanowire@NiCo_2_O_4_ nanosheet array were 2642 F g^−1^, 2473 F g^−1^, 2234 F g^−1^, and 1905 F g^−1^. With the increase in scan speeds, the specific capacitance decreased gradually (Figure 5c). The galvanostatic charge–discharge curves of the samples were tested at different current densities (2.5, 5, 10, and 20 A g^−1^) in the voltage range of −0.1~0.45 V. The specific capacitance of constant current charge–discharge curves can be calculated according to the above Equation (2). Figure 5b shows the galvanostatic charge–discharge curves of the Co_3_O_4_ nanowire array. When the current densities were 2.5 A g^−1^, 5 A g^−1^, 10 A g^−1^, and 20 A g^−1^, the corresponding specific capacitance were 1112 F g^−1^, 982 F g^−1^, 834 F g^−1^, and 680 F g^−1^. Figure 5d shows the galvanostatic charge–discharge curves of the Co_3_O_4_ nanowire@NiCo_2_O_4_ nanosheet hierarchical array. When the current densities were 2.5 A g^−1^, 5 A g^−1^, 10 A g^−1^, and 20 A g^−1^, the specific capacitance values were 2034 F g^−1^, 1880 F g^−1^, 1688 F g^−1^, and 1339 F g^−1^, respectively (Figure 5d).

Figure 6a,b shows the comparison of cyclic voltammetry curves of the Co_3_O_4_ nanowire array and the Co_3_O_4_ nanowire@NiCo_2_O_4_ nanosheet hierarchical array at a current density of 25 mV s^−1^ and the galvanostatic charge–discharge comparison at a current density of 5 A g^−1^. The specific capacitance of the Co_3_O_4_ nanowire@NiCo_2_O_4_ nanosheet hierarchical array was higher than that of the Co_3_O_4_ nanowire, which may because the two-dimensional nanosheet has a larger BET and has more contact surfaces during the redox reaction of the electrode than the one-dimensional nanowire. The specific capacitance of the Co_3_O_4_ nanowire@NiCo_2_O_4_ nanosheet hierarchical array was significantly better than that of the Co_3_O_4_-based nanocomposites reported in the literature, such as Co_3_O_4_@MnO_2_ (1532.4 F g^−1^, 1 A g^−1^) [43] and Co_3_O_4_@NiCo_2_O_4_ (672 F g^−1^, 0.5 A g^−1^) [37].

Galvanostatic charge–discharge cycling of the Co_3_O_4_ nanowire array and the Co_3_O_4_ nanowire @NiCo_2_O_4_ nanosheet hierarchical array were conducted 5000 times at a current density of 5 A g^−1^ to study the cycling performance. As shown in Figure 6a, the specific capacitance of the Co_3_O_4_ nanowire array is 91.2%, and that of the Co_3_O_4_ nanowire @NiCo_2_O_4_ nanosheet hierarchical array is 94.7%. A key factor in the practical application of electrode materials is cycling performance. According to the literature report, the following results obtain: mesoporous Co_3_O_4_ ultra-thin nanosheets (78.5%, 2000 cycles) [8]; Co_3_O_4_/MnO_2_ (93.4%, 5000 cycles) [43]; and CoMoO_4_/Co_3_O_4_ (90.38%, 2000 cycles) [39]. The excellent cycling performance of Co_3_O_4_ nanowires@NiCo_2_O_4_ nanosheet hierarchical array electrode materials provided a good theoretical basis for practical applications (Table 1).

The frequency range of the electrochemical impedance was 100 kHz to 0.05 Hz with the open-circuit voltage of 5 mV (Figure 7). Electrochemical impedance was composed of electrolyte ion resistance, the inherent resistance of the active material, and the contact resistance between the active material and the collector. The diameter of the fitting circle of the quasi-semicircle in the high-frequency region represented the charge transfer resistance of the electrode material in the electrochemical system [44,45,46]. As shown in Figure 6b, the resistance of the Co_3_O_4_ nanowire @NiCo_2_O_4_ nanosheet hierarchical array electrode material is significantly lower than that of the Co_3_O_4_ nanowire array. The lower resistance of charge transfer enables the hierarchical array to have a better electrochemical performance [47,48].

## 4. Conclusions

In summary, the Co_3_O_4_ nanowire@NiCo_2_O_4_ nanosheet hierarchical array was constructed on Ni foam via hydrothermal synthesis followed by calcination treatment. The XRD, SEM, and TEM results demonstrated that the NiCo_2_O_4_ nanosheet wrapped on the surfaces of the Co_3_O_4_ nanowire provided faster ion and electron transfer and improved structural integrity. The Co_3_O_4_ nanowire@NiCo_2_O_4_ nanosheet hierarchical array exhibited high specific capacitance and excellent cycling performance. The results showed that the Co_3_O_4_ nanowire@NiCo_2_O_4_ nanosheet hierarchical array would be a desirable electrode material for a high-performance supercapacitor.

## Figures and Tables

**Figure 1 nanomaterials-14-01703-f001:**
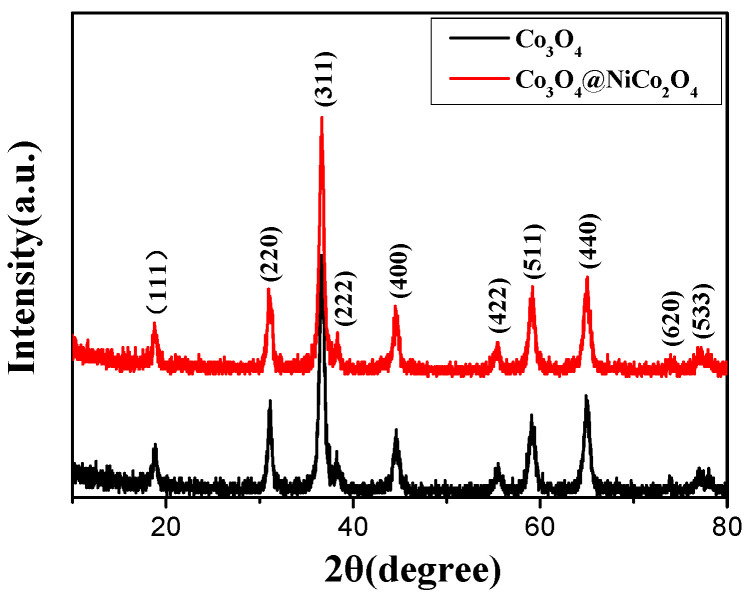
The XRD patterns of the Co_3_O_4_ nanowire and the Co_3_O_4_ nanowire@NiCo_2_O_4_ nanosheet hierarchical array.

**Figure 2 nanomaterials-14-01703-f002:**
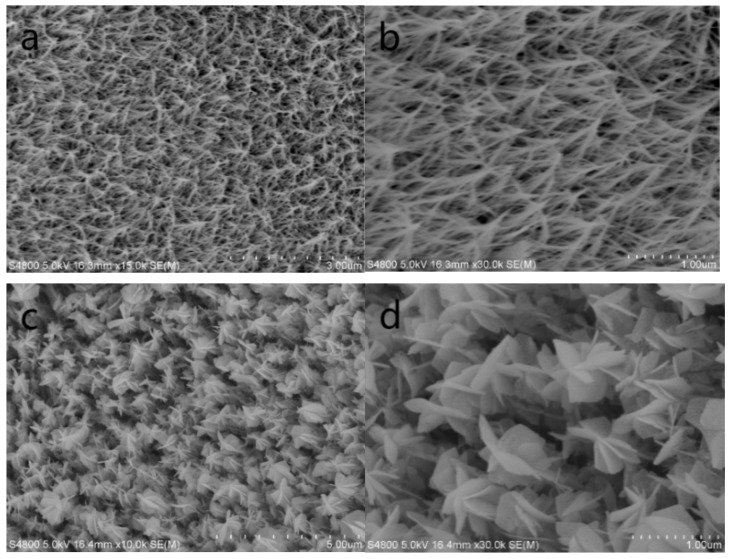
(**a**) Low- and (**b**) high-magnification SEM images of the Co_3_O_4_ nanowire. (**c**) Low- and (**d**) high-magnification SEM images of the Co_3_O_4_ nanowire@NiCo_2_O_4_ nanosheet hierarchical array.

**Figure 3 nanomaterials-14-01703-f003:**
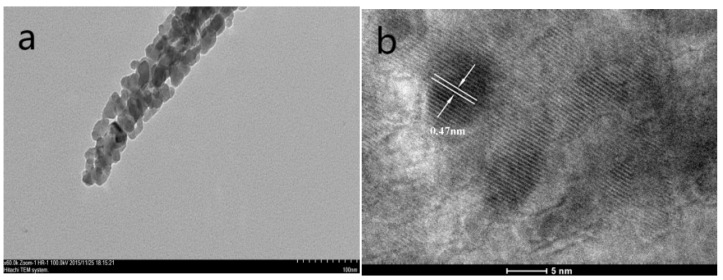
(**a**) TEM and (**b**) HRTEM images of the Co_3_O_4_ nanowire. (**c**) TEM and (**d**) HRTEM images of theCo_3_O_4_ nanowire@NiCo_2_O_4_ nanosheet hierarchical array.

**Figure 4 nanomaterials-14-01703-f004:**
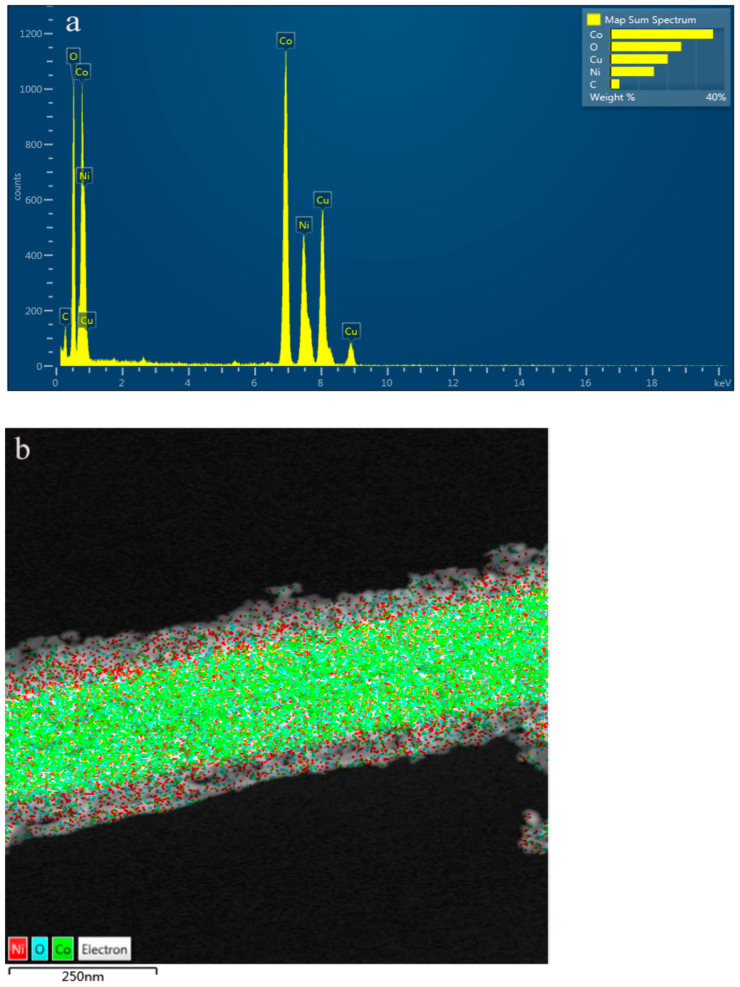
(**a**) Element proportion of the Co_3_O_4_ nanowire@NiCo_2_O_4_ nanosheet and (**b**) EDS image of the Co_3_O_4_ nanowire@NiCo_2_O_4_ nanosheet.

**Figure 5 nanomaterials-14-01703-f005:**
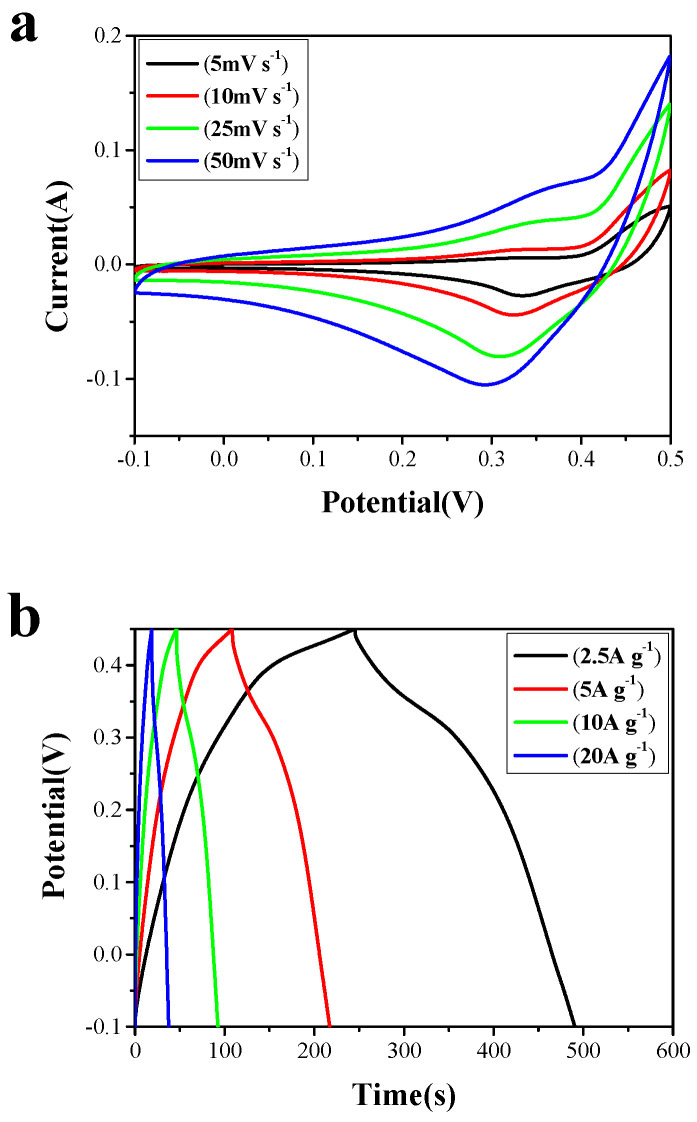
(**a**) Cyclic voltammetry curves of the Co_3_O_4_ nanowire and (**b**) galvanostatic charge–discharge curves of Co_3_O_4_ nanowire. (**c**) Cyclic voltammetry curves of the Co_3_O_4_ nanowire@NiCo_2_O_4_ nanosheet and (**d**) galvanostatic charge–discharge curves of the Co_3_O_4_ nanowire@NiCo_2_O_4_ nanosheet.

**Figure 6 nanomaterials-14-01703-f006:**
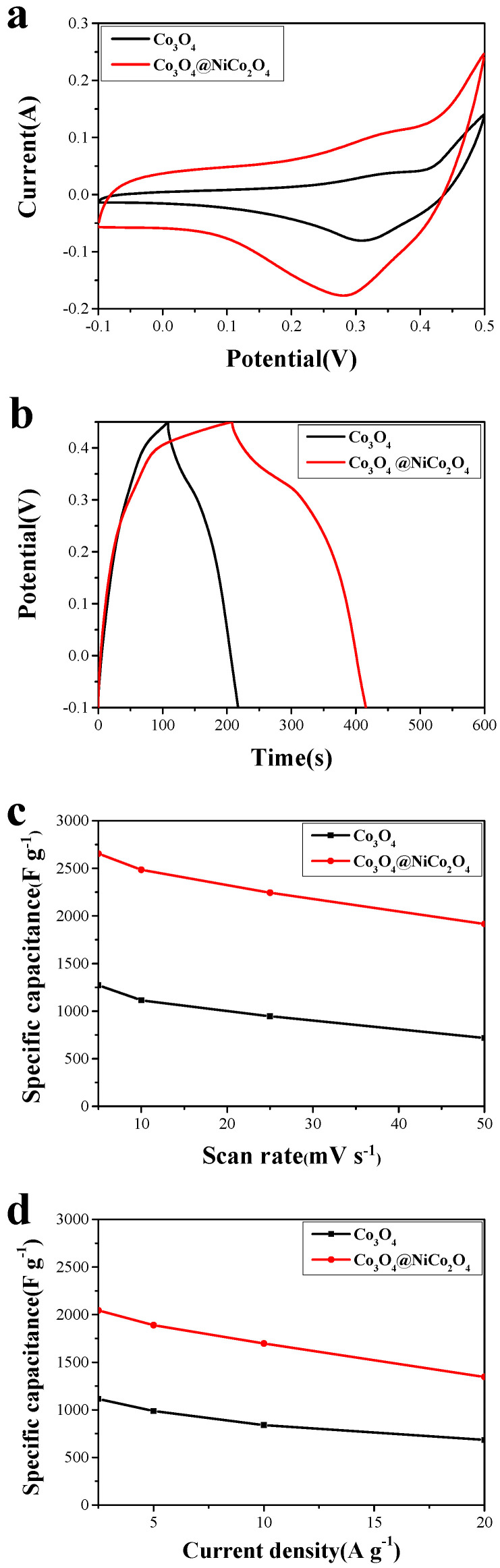
The electrochemical performances comparison of the samples: (**a**) the CVs at 25 mV s^−1^; (**b**) the charge–discharge curves at 5 A g^−1^; (**c**) the specific capacitance as a function of scan rate; (**d**) the specific capacitance as a function of current density.

**Figure 7 nanomaterials-14-01703-f007:**
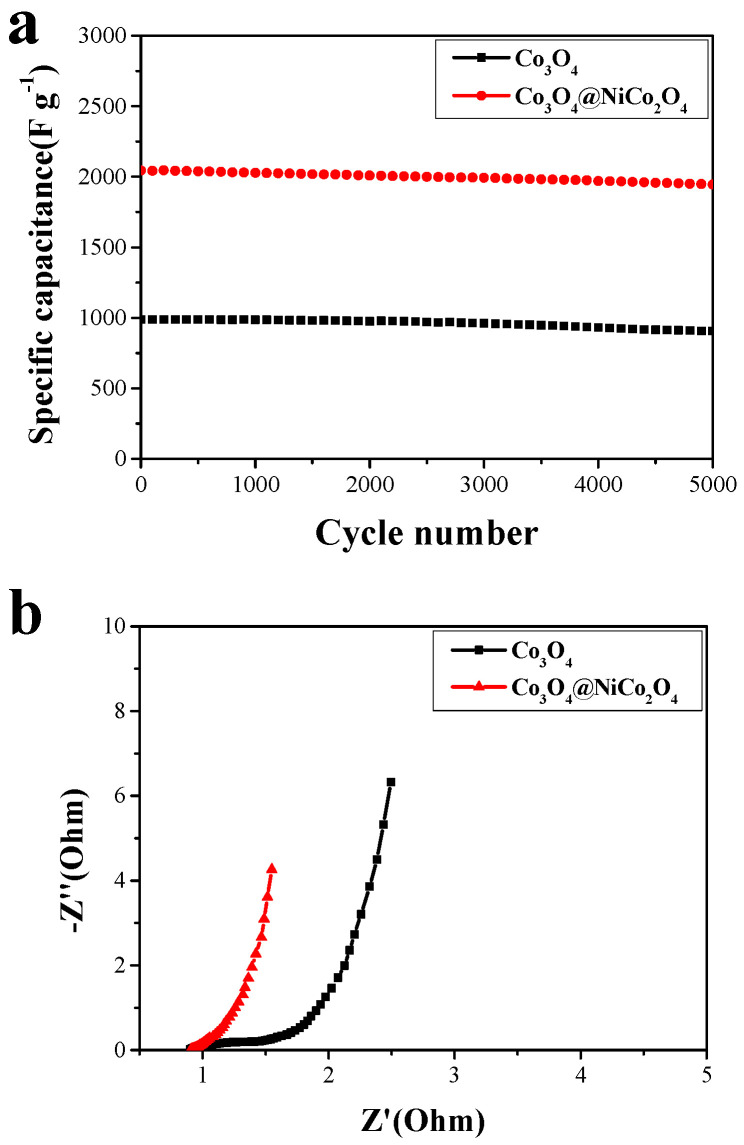
(**a**) The cyclic performance of charge–discharge for 5000 cycles at 5 A g^−1^. (**b**) Nyquist plots at 1 A g^−1^.

**Table 1 nanomaterials-14-01703-t001:** Comparison of cycling performance.

Materials	Capacitance Remaining	Cycles
Co_3_O_4_ nanowire @NiCo_2_O_4_ nanosheet	94.7%	5000
Co_3_O_4_ nanowire	91.2%	5000
mesoporous Co_3_O_4_ ultra-thin nanosheets	78.5%	2000
Co_3_O_4_/MnO_2_	93.4%	5000
CoMoO_4_/Co_3_O_4_	90.38%	2000

## Data Availability

Data are contained within the article.

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
