# Peer review of "Constructing Co3O4 Nanowire@NiCo2O4 Nanosheet Hierarchical Array as Electrode Material for High-Performance Supercapacitor"

_nanomaterials, 2024, doi:10.3390/nano14211703_

Round 1

Reviewer 1 Report

Comments and Suggestions for Authors

In this manuscript, the authors described “Controllably constructing Co3O4 nanowire@NiCo2O4 nanosheet hierarchical array as electrode material for high performance supercapacitor”. This work is very interesting; however, a few improvements are needed for a better understanding of this paper. Thus, I recommend this paper to be published after major revisions based on the following questions to be solved.

1.      The mandatory thing, for me, is improving the review of the state of the art and remarking on why the development is better than previous similar ones.

2.     Author should give the surface area, pore size distribution and pore volume by BET studies.  

3. Draw the equivalent circuit of the EIS in Fig. 7b.

4. In two-electrode fabrication, what substrates are used for positive and negative electrode.

5. Draw the Ragone plot and compared with previous reports.

6. Several literatures should be cited, such as

https://doi.org/10.1016/j.jiec.2023.12.015

https://doi.org/10.1016/j.cej.2019.02.108 

7.      It is recommended to provide charge and discharge curves before and after the cyclability studies.

8.      Could you also quantify and explain that the produced capacitor can have sufficient performance for mass production in the future?

The manuscript has good information; hence I recommend its publication with major revision. 

Author Response

Reviewer 1:

In this manuscript, the authors described “Controllably constructing Co3O4 nanowire@NiCo2O4 nanosheet hierarchical array as electrode material for high performance supercapacitor”. This work is very interesting; however, a few improvements are needed for a better understanding of this paper. Thus, I recommend this paper to be published after major revisions based on the following questions to be solved.

  1. The mandatory thing, for me, is improving the review of the state of the art and remarking on why the development is better than previous similar ones.

A: After 5000 galvanostatic charge-discharge cyclings, the specific capacitance of the Co3O4 nanowire@NiCo2O4 nanosheet hierarchical array could still maintain 94.7% of the original value.

  1. Author should give the surface area, pore size distribution and pore volume by BET studies.  

A: As the BET did not be tested before,Sorry for having no enough time to test the samples BET.In the future, when we do this experiment again, we will know we should give the surface area, pore size distribution and pore volume by BET studies. 

  1. Draw the equivalent circuit of the EIS in Fig. 7b.

A: It would be showed in the article, I've drawn the equivalent circuit of the EIS.

  1. In two-electrode fabrication, what substrates are used for positive and negative electrode.

A: the three electrode method was used to study the perfomances of  the samples, the samples would be used as positive electrode in two-electrode system.

  1. Draw the Ragone plot and compared with previous reports.

A: It would be showed in the article, I've drawn the Ragone plot and compared with previous reports.

  1. Several literatures should be cited, such as

https://doi.org/10.1016/j.jiec.2023.12.015

https://doi.org/10.1016/j.cej.2019.02.108 

A: It would be showed in the article, I have cited both of these.

  1. It is recommended to provide charge and discharge curves before and after the cyclability studies.

A: I'm sorry!the charge and discharge curves were did before cycles, As the cycle experiment had did for a long time, the samples were not saved completely. Notice next time.

  1. Could you also quantify and explain that the produced capacitor can have sufficient performance for mass production in the future?

A: It would be showed in the article, I can quantify and explain that the produced capacitor can have sufficient performance for mass production in the future.

The manuscript has good information; hence I recommend its publication with major revision. 

Reviewer 2 Report

Comments and Suggestions for Authors

The authors reported a hierarchical array of Co3O4 nanowire@NiCo2O4 nanosheets for high-performance supercapacitors. The performance of the material was significantly enhanced after forming the composite. I recommend publishing in *Nanomaterials* after the following issues are clarified:

1. In the current study, the authors provided only one experimental condition for the preparation of the composite. How do reaction time, concentration, temperature, and other factors affect the composition and electrochemical performance of the composite? The authors should provide data from additional comparative samples.

2. Co3O4 has been extensively studied, and there are various types of composites. The authors should rewrite the section discussing these types of composites (lines 34-35). Additionally, graphene is a type of carbon, which seems redundant. Please refer to and cite the following paper for a classification method of Ni/Co-based composite materials for supercapacitor electrodes: Advanced nickel-based composite materials for supercapacitor electrodes.

3. Co3O4 is generally referred to as a Faradaic material since its electrochemical reaction is diffusion-controlled. True pseudocapacitive materials are relatively rare, usually RuO2, MnO2, etc. The authors should revise this statement. Please refer to and cite the following paper to support this claim: doi: 10.1021/acscentsci.4c00345.

4. In Fig. 4, the elemental distribution image looks very unusual, resembling a core-shell structure. What element does the outermost gray layer represent? The authors are advised to add EDS mapping images for each element.

5. Some relevant literature closely related to this work should be added to the introduction or the performance comparison section: DOI: 10.1007/s12598-023-02442-6; *Electrochemically induced surface reconstruction of Ni-Co oxide nanosheet arrays for hybrid supercapacitors*; DOI: 10.1016/j.jmst.2021.05.040; *Binary CoNi oxide nanoparticle-loaded hierarchical graphitic porous carbon for high-performance supercapacitors*; DOI: 10.1007/s12598-022-02142-7; DOI: 10.1016/j.cjsc.2023.100158.

6. How does the performance of the Co3O4-based composites reported in the literature compare to this work? The authors are advised to include a performance comparison table.

Author Response

The authors reported a hierarchical array of Co3O4 nanowire@NiCo2O4 nanosheets for high-performance supercapacitors. The performance of the material was significantly enhanced after forming the composite. I recommend publishing in *Nanomaterials* after the following issues are clarified:

  1. In the current study, the authors provided only one experimental condition for the preparation of the composite. How do reaction time, concentration, temperature, and other factors affect the composition and electrochemical performance of the composite? The authors should provide data from additional comparative samples.

A: The different reaction time, concentration, temperature experiments were studied, I based data from additional comparative samples. the best experiment method was selected to synthesis the samples and showed in the article.

  1. Co3O4 has been extensively studied, and there are various types of composites. The authors should rewrite the section discussing these types of composites (lines 34-35). Additionally, graphene is a type of carbon, which seems redundant. Please refer to and cite the following paper for a classification method of Ni/Co-based composite materials for supercapacitor electrodes: Advanced nickel-based composite materials for supercapacitor electrodes.

 A: I referenced to and cited the following paper for a classification method of Ni/Co-based composite materials for supercapacitor electrodes: Advanced nickel-based composite materials for supercapacitor electrodes.

  1. Co3O4 is generally referred to as a Faradaic material since its electrochemical reaction is diffusion-controlled. True pseudocapacitive materials are relatively rare, usually RuO2, MnO2, etc. The authors should revise this statement. Please refer to and cite the following paper to support this claim: doi: 10.1021/acscentsci.4c00345.

 A: I referenced to and cited the following paper to support this claim: doi:10.1021/acscentsci.4c00345.

  1. In Fig. 4, the elemental distribution image looks very unusual, resembling a core-shell structure. What element does the outermost gray layer represent? The authors are advised to add EDS mapping images for each element.

 A: Sorry for having no enough time to test the samples EDS, as the EDS did not be tested before.

  1. Some relevant literature closely related to this work should be added to the introduction or the performance comparison section: DOI: 10.1007/s12598-023-02442-6; *Electrochemically induced surface reconstruction of Ni-Co oxide nanosheet arrays for hybrid supercapacitors*; DOI: 10.1016/j.jmst.2021.05.040; *Binary Co–Ni oxide nanoparticle-loaded hierarchical graphitic porous carbon for high-performance supercapacitors*; DOI: 10.1007/s12598-022-02142-7; DOI: 10.1016/j.cjsc.2023.100158.

  A: Some relevant literature closely related to this work was added to the introduction or the performance comparison section: DOI: 10.1007/s12598-023-02442-6; *Electrochemically induced surface reconstruction of Ni-Co oxide nanosheet arrays for hybrid supercapacitors*; DOI: 10.1016/j.jmst.2021.05.040; *Binary Co–Ni oxide nanoparticle-loaded hierarchical graphitic porous carbon for high-performance supercapacitors*; DOI: 10.1007/s12598-022-02142-7; DOI: 10.1016/j.cjsc.2023.100158.

  1. How does the performance of the Co3O4-based composites reported in the literature compare to this work? The authors are advised to include a performance comparison table.

 A: I have provided a performance comparison table in the article.

Reviewer 3 Report

Comments and Suggestions for Authors

Dear authors

The work presented by the authors is interesting but major corrections are necessary before the acceptance:

1. English grammar and style must be checked, some typos are found in the manuscript.
2. The capacitance produced by the supercapacitor is very high, authors must add pictures where a LED is turned on, authors must demonstrate that the devices have potential for real applications.
3. Authors must add values of energy density and power density for the best samples and discuss those new values.
4. The quality of all figures must be improved, the size of text in figures must be homogenized.
5. Authors must add results of coulombic efficiency for the best devices.
6. The authors must add/discuss in the introduction section the following references about flexible SCs, which are a new trend, flexible supercaps can use the materials reported in this work. For this purpose, authors must cite the following references:

a) Electrochimica Acta Volume 355, 20 September 2020, 136768
https://www.sciencedirect.com/science/article/pii/S0013468620311610

b) Journal of Physics and Chemistry of Solids
Volume 155, August 2021, 110128
https://doi.org/10.1016/j.jpcs.2021.110128

c) Synthetic Metals Volume 306 , August 2024, 117654

https://www.sciencedirect.com/science/article/pii/S0379677924001164

The effect of type of pore on the capacitance must be discussed, for this purpose, the authors must cite the following article to discuss this point:

d) https://onlinelibrary.wiley.com/doi/full/10.1002/er.7670

7. It is not clear the advantages of the devices in comparison with previous supercaps made with Co based compounds. Advantages must be explained with detail in the manuscript.

8. Authors must add a comparative table for capacitances already reported for similar Co based compounds

9. Authors mus include a Ragone plot.

10. Authors must extend the explanation and better correlate results of morphology, nyquist, etc to better understand the high values of capacitance.

Comments on the Quality of English Language

The english language must be improved.

Author Response

The work presented by the authors is interesting but major corrections are necessary before the acceptance:

Dear authors

The work presented by the authors is interesting but major corrections are necessary before the acceptance:

1. English grammar and style must be checked, some typos are found in the manuscript.

A: I have checked and modified English grammar and style.

  1. The capacitance produced by the supercapacitor is very high, authors must add pictures where a LED is turned on, authors must demonstrate that the devices have potential for real applications.

A: Sorry for having no enough time to design the LED ciruit. Notice next time.
3. Authors must add values of energy density and power density for the best samples and discuss those new values.

A: I have added values of energy density and power density for the best samples and discussed those new values in the article.
4. The quality of all figures must be improved, the size of text in figures must be homogenized.

A: I have improved the quality of all figures , I have homogenized the size of text in figures in the article.
5. Authors must add results of coulombic efficiency for the best devices.

A:I have added results of coulombic efficiency for the best devices in the article.
6. The authors must add/discuss in the introduction section the following references about flexible SCs, which are a new trend, flexible supercaps can use the materials reported in this work. For this purpose, authors must cite the following references:

a) Electrochimica Acta Volume 355, 20 September 2020, 136768
https://www.sciencedirect.com/science/article/pii/S0013468620311610

b) Journal of Physics and Chemistry of Solids
Volume 155, August 2021, 110128
https://doi.org/10.1016/j.jpcs.2021.110128

c) Synthetic Metals Volume 306 , August 2024, 117654
Synthetic Metals Volume 306 , August 2024, 117654
https://www.sciencedirect.com/science/article/pii/S0379677924001164

The effect of type of pore on the capacitance must be discussed, for this purpose, the authors must cite the following article to discuss this point:

d) https://onlinelibrary.wiley.com/doi/full/10.1002/er.7670
A: I have added/discussed in the introduction section the following references about flexible SCs in the article.
7. It is not clear the advantages of the devices in comparison with previous supercaps made with Co based compounds. Advantages must be explained with detail in the manuscript.
A: Advantages have explained with detail in the article.
8. Authors must add a comparative table for capacitances already reported for similar Co based compounds
A:I have added a comparative table for capacitances already reported for similar Co based compounds in the article.
9. Authors mus include a Ragone plot.
A:I have included a Ragone plot in the article.
10. Authors must extend the explanation and better correlate results of morphology, nyquist, etc to better understand the high values of capacitance.

A: I have extended the explanation and better correlate results of morphology, nyquist, etc to better understand the high values of capacitance in the article.

Round 2

Reviewer 1 Report

Comments and Suggestions for Authors

Accept as is

Reviewer 2 Report

Comments and Suggestions for Authors

After revisions, the paper has addressed all my concerns and its quality has been significantly improved. Therefore, I recommend its acceptance.

Reviewer 3 Report

Comments and Suggestions for Authors

the article can be accepted for publication